# New estimates indicate that males are not larger than females in most mammal species

Kaia J. Tombak [1,2] ✉, Severine B. S. W. Hex [2] & Daniel I. Rubenstein [2]

Sexual size dimorphism has motivated a large body of research on mammalian mating strategies and sexual selection. Despite some contrary evidence, the narrative that larger males are the norm in mammals—upheld since Darwin's *Descent of Man*—still dominates today, supported by meta-analyses that use coarse measures of dimorphism and taxonomically-biased sampling. With newly-available datasets and primary sources reporting sex-segregated means and variances in adult body mass, we estimate statistically-determined rates of sexual size dimorphism in mammals, sampling taxa by their species richness at the family level. Our analyses of wild, non-provisioned populations representing >400 species indicate that although males tend to be larger than females when dimorphism occurs, males are not larger in most mammal species, suggesting a need to revisit other assumptions in sexual selection research.

A long-standing narrative postulates that in mammals, males are typically larger than females. Darwin treated it as a matter of common knowledge[1], as have many subsequent evolutionary biologists studying sexual selection[2-7]. The principal hypothesis predicting a prevalence of larger males in mammals is that the reproductive investment that females commit to their offspring (via gestation, lactation, and often parental care) results in a skewed operational sex ratio, leading to greater mate competition and selection for competitive ability among males[1,8,9]. This pattern should be especially strong under polygyny, presumed to be the most common mating system in mammals[8,10]. In the 1970's, Ralls contributed the first review of rates of sexual size dimorphism (SSD) in mammals and found weak support for this hypothesis. She concluded that most mammals are 'not extremely dimorphic', that species with little sexual size dimorphism were extremely numerous in the most species-rich mammalian orders[11], and that larger females were surprisingly common in mammals[12]. Nonetheless, her findings have been overpowered by the continuation of the 'larger males' narrative[2-7], despite some additional evidence supporting her conclusions[13,14].

Issues with data availability and taxonomic biases have hindered efforts to accurately estimate the rate of SSD in mammals. Meta-analyses have so far been limited to using mean adult body mass values for each sex, a measure that is widely available in the literature but typically reported without measures of variance that would allow for a statistical assessment of dimorphism. To designate species as dimorphic or monomorphic, researchers have therefore used either arbitrary cut-offs (including a 5%[15], 10%[5], and up to a 20% difference in mean body mass[16]) or ratios below or above 1 between mean male and female body masses[5,17]. Ralls herself used mean body mass ratios to assess the modal degree of dimorphism in each mammalian order[11]. Using different criteria influences the conclusions. For instance, Lindenfors et al. [5] concluded that mammals generally had male-biased size dimorphism because the average body mass ratio was >1 across their sample, but their other analyses using a 10% body mass difference cut-off indicated that less than half of mammalian species had male-biased SSD[5]. Neither criterion is based on sufficient information to determine rates of dimorphism: body mass difference thresholds are both arbitrary and inconsistent, and an average body mass ratio >1 across species can indicate either more frequent dimorphism or more extreme dimorphism in one sex than the other. In addition, research on SSD in mammals has tended to focus on a few taxa, namely artiodactyls, carnivores (especially pinnipeds), and primates[8,17-21]: clades with high rates of male-biased SSD[22]. However, most mammals, by far, are rodents and bats[23], which are often under-represented in studies of SSD. The phylogenetic signal for SSD is strong[24], calling for updated estimates with more balanced taxonomic representation.

[1]Department of Anthropology, Hunter College of the City University of New York, New York, NY, USA. [2]Department of Ecology and Evolutionary Biology, Princeton University, Princeton, NJ, USA. ✉e-mail: ktombak@alumni.princeton.edu

Fortunately, some recently-published large datasets report mean body mass as well as measures of variance for each sex across mammalian taxa. We combined these datasets with data from primary sources to revisit Ralls's original question, estimating the rates of sexual size dimorphism in wild, non-provisioned mammalian populations using statistical determinations of dimorphism for each species and sampling each mammalian order and family according to their species richness.

## Results

Our final dataset included 429 species with a minimum sample size of 9 for each sex: the minimum sample size that mitigates for the inflation of confidence intervals with low sample size in our dataset (Supplementary Fig. 1). We achieved at least 5% representation for each mammalian order except for Eulipotyphla (3.8%). We also achieved at least 5% representation for 66 out of the 78 mammalian families comprising at least 10 species (Supplementary Fig. 2, Supplementary Table 1). Our estimates, based on the frequency with which the 95% confidence interval for the between-sex difference in mean body mass straddles zero, and weighted by species richness in each family, indicated that 38.7% of mammalian species are sexually monomorphic in body mass, while 45.1% of species are male-biased dimorphic and 16.2% are female-biased dimorphic (Fig. 1).

Male-biased dimorphism was somewhat more extreme on average than female-biased dimorphism (mean male/female body mass ratio in male-biased dimorphic species = 1.28, $N = 178$; mean female/male body mass ratio in female-biased dimorphic species = 1.13, $N = 71$). This confirms that average male/female mass ratios >1 are inappropriate indicators of the frequency of dimorphism. The most dimorphic species was the northern elephant seal (*Mirounga angustirostris*), where males had a mean mass 3.2 times that of females[25]. The most extreme female-biased dimorphism was found in the peninsular tube-nosed bat (*Murina peninsularis*), in which mean female mass was 1.4 times that of males[26]. However, most dimorphisms were not extreme (Fig. 2), as Ralls concluded almost 50 years ago[11]. When we reran the analyses on rates of SSD on body length instead of body mass in the subset of our data with body length measurements (see Methods), our estimates shifted towards more monomorphism

and female-biased dimorphism (49.9% monomorphic, 28.0% male-biased dimorphic, 22.1% female-biased dimorphic; $N = 199$).

Overall, standard deviation in body mass was greater in males (mean SD = 1980.6 g, median SD = 12.2 g) than in females (mean SD = 1200.2 g, median SD = 10.7 g), so our results were unlikely to be seriously confounded by data that may have included pregnant females without our knowledge (paired Wilcoxon Rank Sum test V = 58215, $p < 0.0001$). Further, standard deviation in body mass was greater in males than females among male-biased dimorphic species (median male SD = 47.9 g, median female SD = 36.6 g, V = 13801, $p < 0.0001$), greater in females than males in female-biased dimorphic species (median male SD = 2.5 g, median female SD = 3.0 g, V = 536, $p < 0.0001$), and no different between the sexes in monomorphic species (median male SD = 8.4 g, median female SD = 9.3 g, V = 7473, $p = 0.34$).

Patterns of SSD differed markedly between orders (Figs. 1 and 2, Supplementary Fig. 2). About half of the species in Rodentia (the most species-rich order) were monomorphic, whereas close to half of Chiroptera (the second-most species-rich order) had larger females (Supplementary Table 1). Larger males were the norm for several of the less species-rich orders, while several others were evenly divided between larger males and monomorphism, and larger females were the norm for Lagomorpha (Fig. 2, Supplementary Fig. 2). Notably, the orders that had the most prevalent male-biased dimorphism included Artiodactyla, Carnivora, and Primates: the orders that dominate the SSD literature for mammals[8,17–21]. Differences in rates of SSD at the family level were also evident, indicating that weighting our estimates based on species richness in each family was important. For example, the famously larger females in Lagomorpha were so only in Leporidae (rabbits and hares), while Ochotonidae (pikas) have monomorphism and male-biased dimorphism (Supplementary Fig. 2). In Primates, larger males are the norm overall, but strepsirrhine primates are mostly monomorphic, as are about half of Cebidae (Supplementary Fig. 2, Supplementary Table 1).

## Discussion

Our results did not support the 'larger males' narrative−the idea that most mammals have larger males than females. While species with

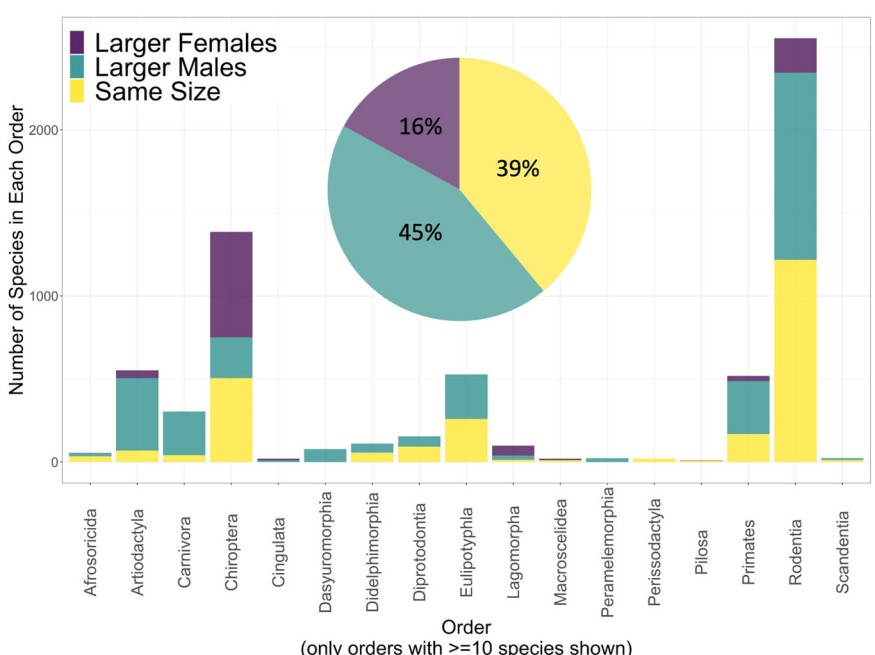

**Fig. 1 | Estimated rates of sexual dimorphism in body mass in mammals.** Rates are displayed for mammals as a whole (pie chart), and for each mammalian order comprising at least ten species (bar chart), and all estimates are weighted by species richness in each family.

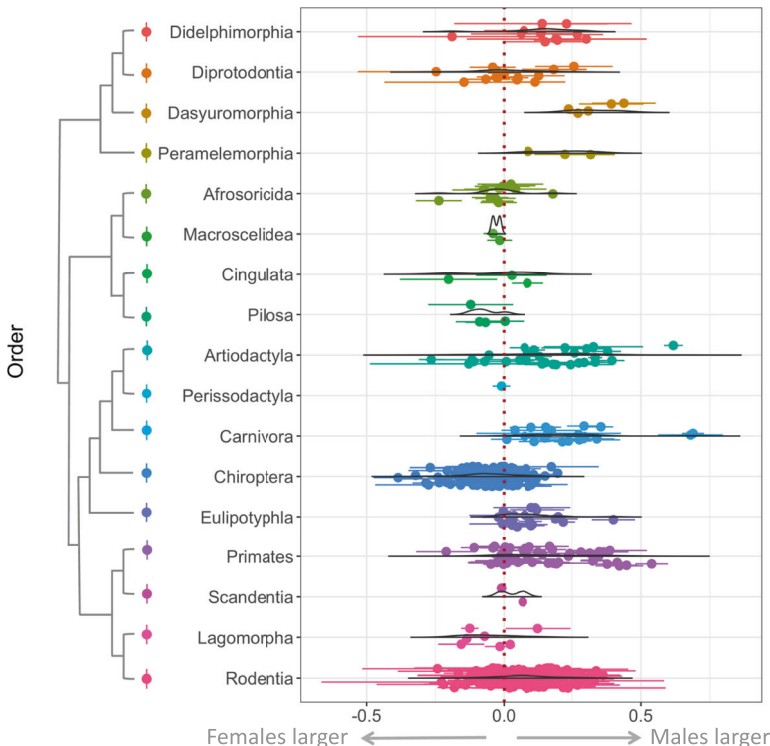

**Fig. 2 | Distribution of mean body mass differences between the sexes in mammals.** Mean mass differences are displayed relative to male body mass for all data included in analyses, color-coded by order. A proportional mass difference of −0.2, for example, indicates that females are 20% larger than males on average. Error bars indicate 95% confidence intervals, data density distributions are scaled such that the area under the curve is the same for each order, and the phylogeny on the left is derived from Upham et al. [87]. Please see Supplementary Table 1 for the number of species sampled for each order and family, as well as the full datasheet on Dryad for the number of individuals sampled for each species.

larger males were the largest single category, we found that males are not larger than females in most mammalian species, and that sexual size monomorphism was almost as frequent as larger males (and potentially more frequent if body length is used as the measure of size instead of body mass).

Importantly, ours should not be the last word on rates of sexual size dimorphism in mammals. First, we prioritized data quality over quantity, and our conclusions are based on data covering only 5% of mammalian species. However, our results align well with those based on Lindenfors et al.'s estimates using a cruder criterion for dimorphism across >1300 mammal species[5]. Second, some minor taxonomic biases persist in our dataset. Mammals of very high body mass are difficult to weigh and, when such data are reported, often have low sample sizes. However, the most underrepresented taxa by species richness were still small-bodied, speciose clades (certain families in Eulipotyphla, Chiroptera, and Rodentia; Supplementary Fig. 2). Third, given the suppressed reporting of non-significant results in science (and we frequently saw statements of a lack of sexual dimorphism unaccompanied by descriptive statistics in primary sources)[27], our estimated rates of monomorphism are probably underestimates. Finally, body mass varies by body condition and is not an ideal measure of size for many taxa[13,28,29]. Still, our preliminary results showing a predominance of sexual monomorphism in body length in mammals reinforce the idea that it may be time to retire the 'larger males' narrative. All in all, our results accord with Ralls's original reviews[11,12], with smaller-scale meta-analyses on species-rich mammalian taxa[13,30], and with Lindenfors et al.'s large-scale meta-analysis that found the same rate of male-biased SSD as we did using a 10% body mass difference cut-off[5]. Yet the latter study fell back on male/female body mass ratios to conclude that mammals generally have larger males and Ralls's

review—which was the first review of the evidence on rates of SSD in mammals—has been miscited several times as having supported the 'larger males' narrative[5,7,24].

Why has this narrative persisted so stubbornly? It may be ascribed to the long-time focus of SSD research on species with conspicuous dimorphisms, as suggested by Bondrup-Nielsen and Ims[31] and by Dewsbury et al.[32]. However, given the well-established variation in dimorphism across mammalian taxa, it is surprising that so many would accept generalizations based on a few, relatively species-poor taxa. The narrative may also be traced to a long-standing research focus on male mating strategies in the study of evolution[33,34], particularly in mammals[35]. Darwin himself focused almost entirely on how sexual selection operated on males in the form of mate competition when discussing mammals[1]. Competitive males and choosy females are a recurring theme in animal behavior research[34], based on the argument that females invest more energy in gametes and are therefore the less reproductively available sex: the controversial 'Darwin-Bateman-Trivers' paradigm[1,2,36]. The dominance of this paradigm and the general focus on males in sexual selection research are likely to have influenced which narratives are readily accepted and amplified and which are overlooked or subjected to heavier scrutiny[34,37–39].

Shifting the framework of sexual selection research away from the presumption of larger males opens up a set of interesting questions for future investigation and for the advancement of theory. If females must invest disproportionately into growing and raising offspring in mammals in particular, why are so many mammals monomorphic in body size? This is not easily explained by monogamy, which is thought to occur in relatively few mammal species and does not necessarily obviate the need to demonstrate individual quality in mate competition[10,11]. Existing theory for explaining monomorphism or

female-biased dimorphism ranges from clade-specific to generalizable. For example, greater size in female bats has been argued to facilitate carrying embryos and offspring in flight[7,40]. Ralls[12] put forward the Big Mother Hypothesis—the idea that larger females may be better mothers, more capable of providing homeothermy during pregnancy, good-quality milk, protection, transport, and other forms of parental care. This has been suggested to contribute to female-biased SSD in chipmunks (*Tamias* spp.)[29]. However, whether there is a fecundity advantage, disadvantage, or neither for larger females in mammals is controversial[6,13,41]. Mechanisms of sexual selection on males can also take many forms and these have been more thoroughly investigated. For some clades of size-monomorphic rodents, olfactory signaling has been suggested as the trait under sexual selection, rather than size[13,42]. In addition, selection for male agility in combat, rather than size, may account for the prevalence of sexual size monomorphism in equids[43] and in pinnipeds that mate aquatically[44,45] and of smaller males in bats[46]. Finally, sexual selection may occur at precopulatory or post-copulatory stages, and prominent sperm competition should weaken selection for larger male body size, as has been noted in several mammalian clades (e.g., primates[47], voles[48], ungulates[49], cetaceans[50]).

More nuanced investigation into dimorphism would improve sexual selection research, as many species do not fit neatly into a dimorphic or monomorphic category[51]. There can be great intraspecific variation in both body size and dimorphism in mammals. This can come about from temporal variation in body mass at the individual level; for example, extreme seasonal body mass fluctuations in both male and female prairie dogs, *Cynomys* spp., result in males being much larger than females in the beginning of the breeding season but statistically the same size by the end of it[52]. Intraspecific variation in SSD can also come about from variation at the population level, such as a latitudinal cline in the short-nosed fruit bat (*Cynopterus sphinx*) ranging from female-biased to male-biased dimorphism from southern to central India[53], as well as altitudinal variation in SSD in bank voles (*Clethrionomys glareolus*) ranging from female-biased at lower altitudes to male-biased in alpine habitat[54]. High variance in body mass within a sex also presents complications for categorizing species as simply dimorphic or monomorphic; for example, the fossa (*Cryptoprocta ferox*) has two male morphs, one of which is the same size as females and the other of which is larger[55]. Our results indicate that the sex that is larger on average tends to be the sex with greater variance in body mass, challenging the notion that absolute size is strongly linked to sex in most mammals (i.e., a greater average body mass may be driven by a relatively small subset of individuals of the larger sex in many cases). This variety also underlines the potential for multiple alternative reproductive strategies in either sex in mammals[6,51].

Given the building evidence for a greater prevalence of sexual size monomorphism than is commonly recognized, the theoretical basis for the evolution of SSD in mammals deserves some reframing. Reproductive skew is expected to be greater in the more reproductively available sex[56], which in mammals is presumed to be the male most of the time and to result in high mate competition among males. However, a recent review of genetic paternity in mammals found relatively low variance between adult males, indicating more evenly-distributed male reproductive success than expected, and no relationship between paternity variance and sexual size dimorphism across species (including strikingly low reproductive skew among male northern elephant seals)[57]. Multiple reproductive strategies among males may account for this low skew—for example, if there is a trade-off between body size and mortality, larger males may gain more fertilizations over the short-term but smaller males may be more or equally successful over the long-term[9]. How sexual selection, and mate competition in particular, operate among female mammals is generally understudied and deserves greater attention[34]. In animal behavior research, it is commonly assumed that all available females will choose the strongest, most dominant male as a mate, or else be coerced by

him into copulation. However, many populations have shown great variation in female mate preferences[58,59], as well as aggressive competition among females for mates[60,61] and for resources of consequence to their fitness[62], some instances of which have been passed off as capricious behavior rather than adaptive and strategic[34]. Studies on cryptic female choice also hold great promise in further illuminating sexual selection forces, but such studies on mammals are relatively few despite the idea having been discussed for over a quarter century[34,63]. Finally, SSD research should refocus on how multiple selection pressures act on body size in both sexes, including how both broad sexual selection forces and sex-specific pressures balance with natural selection acting on body size in both males and females (e.g., heat stress, agility, and detectability by predators)[6,29,64]. As old assumptions are revisited with larger datasets and greater scrutiny, we see great potential in new breakthroughs in sexual selection theory.

## Methods
### Data collection
We searched Google Scholar between June 2021 and December 2023 for datasets with sex-segregated body mass data for mammalian species that reported means and measures of variance, standard deviation, or 95% confidence intervals as well as sample size for each sex within a population. Extremely large datasets on mean body mass values are available but do not report measures of variance for each sex (e.g., the PanTHERIA database, the Handbook of Mammalian Body Masses, the Handbook of Mammals of the World, AnimalTraits)[65–67]. Many others do not report body mass data in a sex-segregated format (e.g., the Malagasy Animal trait Data Archive, EltonTraits1.0)[68,69]. Some sources combined data from laboratory and wild animals and could not be used[32]. However, we found several published datasets that included data that met our criteria[14,29,31,70–74]. Using these data, we determined that a 5% representation goal was likely feasible across mammalian orders and families. We then searched Google Scholar for primary sources to top up sample sizes for underrepresented orders and families in our dataset, using Burgin et al.'s[23] estimates of species richness in each mammalian order. For primates, we found additional data on underrepresented taxa from the *All the World's Primates* online database[75]. When we found more than one study to report body mass values for the same species, we used the data from the study with greater sample sizes in analyses. Subspecies were treated as different populations of the same species and these data were not combined even if reported in the same study. Our specific goal was to achieve 5% representation of the extant species in every mammalian order with at least 10 species (the lowest number for which 5% can be rounded up to 1 species), and we further aimed to sample 5% of all families with at least 10 species within each of these orders to minimize taxonomic biases.

We excluded any measures from sexually immature, pregnant, or captive-bred animals when these were distinguished (or when it was noted that these were mixed in with the data). Exceptions for inclusion were made for weight measurements from Fischer's pygmy fruit bats (*Haplonycteris fischeri*) for which weights were reported for females with very early-stage fetuses (1.5–5 mm in length), for data from Hottentot golden moles (*Amblysomus hottentotus*), in which males were so much larger than females that their relative weights were unlikely to be much affected by fetuses, and from datasets on the delectable soft-furred mouse (*Praomys delectorum*), the harvest mouse (*Micromys minutus*), and the giant anteater (*Myrmecophaga tridactyla*), in which very few females were pregnant. In addition, we accepted data from a combined pool of captive-born and wild-caught northern tree shrews (*Tupaia belangeri chinensis*) because it was noted that these groups did not differ in body mass. When data were presented for non-breeding seasons separately, data from breeding seasons were excluded. Only data from wild animals were used, and free-ranging animals that were provisioned with food were excluded. Estimates based on museum

specimens were generally not used except where relative masses were verified with field data from a single population, and domestic species were also excluded, although we did use data from semi-domesticated, free-ranging reindeer (*Rangifer tarandus*). Only direct body mass measurements were included, with the exception of Baird's beaked whales (*Berardius bairdii*), and male northern elephant seals, for which we accepted body mass estimated based on body lengths, girths, and/or ultrasound measurements due to the logistical difficulty of weighing such heavy animals in the wild[25]. Where means, sample sizes, and measures of variance were reported separately for different seasons or sites for one population, these were combined using Baker and Nissim's[76] equation to calculate the standard error for the combined sample. All data used in this study and the associated sources, as well as detailed comments and justifications for any exceptions made to our inclusion criteria, can be found in the datasheet on Dryad (see Data Availability Statement).

We focused our data search on body mass because these data are the most available measure of body size in the literature. However, we did collect body length measurements wherever these were reported for the same population for which we obtained body mass data, to serve as verification that our conclusions would not be very different if body length were used. For these, we used head and body length (excluding the tail, when possible) for most taxa, but forearm length was used for bats, hindfoot length for lagomorphs, and head length for dasyuromorphs and peramelemorphs, according to the conventions and data availability for these taxa.

### Statistical analyses

We calculated the 95% confidence interval of the difference in mean mass between males and females for each species[77] and labeled each species as either monomorphic (95% CI of mass difference straddles zero) or dimorphic (95% CI does not straddle zero). Our initial dataset comprised a total of 691 populations, including 630 unique mammalian species, but this included some populations with a sample size of only 2 for each sex. Lower sample sizes decrease confidence in the mean masses, broadening the 95% confidence intervals and the likelihood of being assigned as monomorphic. We therefore plotted, for each sex in turn, the difference in mean mass between the sexes, divided by mean body mass for that sex, against sample size and found the elbow of the exponential decay function in this relationship ($N = 9.88$ for males, $N = 9.16$ for females) using the *findCutoff* function in the 'KneeArrower' R package[78] (Supplementary Fig. 1). A minimum sample size of 9 for each sex was thus used as a criterion for inclusion in the analyses and only data from the population with the highest sample size for any given species (including among any of its subspecies) was used, producing a final count of 429 species included.

Some orders and families within orders were highly overrepresented relative to their species richness in our final dataset. To ensure that these did not contribute disproportionately to our estimates of rates of dimorphism, we randomly sampled rows in our dataset within each family such that they were sampled exactly according to their species richness. In other words, each family was assigned the number of rows corresponding to 5% of the number of species in the family (rounded to the closest integer), and only this many rows were randomly sampled, without replacement, from the family to produce estimated average rates of dimorphism for each order. This random sampling within mammalian families was performed 1,000 times and the frequencies of female-biased dimorphism, male-biased dimorphism, and monomorphism were tabulated each time to enable the calculation of species-richness-adjusted average frequencies of dimorphism for each order and for mammals as a whole. The order Pilosa includes ten species but is divided into four families, all of which have fewer than ten species; these were represented in the dataset roughly in proportion to their species richness

and were pooled such that only one row for the order was sampled randomly for each iteration.

To generate estimates of rates of dimorphism using body length data, we similarly sampled the subset of species from the final dataset for which the source reported body length measurements ($N = 192$) by species richness. However, since fewer body length data were available than body mass data, we set the goal to 1% representation and sampled by species richness only at the level of the order, rather than for each family. All analyses and data visualizations were performed in R Studio version 4.3.2[79], using the packages 'ggplot2'[80], 'gridExtra'[81], 'dplyr'[82], 'tidyr'[83], 'viridis'[84], 'KneeArrower'[78], and 'see'[85].

### Reporting summary

Further information on research design is available in the Nature Portfolio Reporting Summary linked to this article.

## Data availability

The data generated in this study, as well as the associated list of data sources, are publicly available on Dryad at: https://doi.org/10.5061/dryad.280gb5mx0[86]. A summary table and bar chart of the results for each order and family may be found in the supplementary materials (Supplementary Table 1, Supplementary Fig. 2).

## Code availability

All R code used to process and analyze the data and to generate data visualizations is publicly available on Zenodo at https://doi.org/10.5061/dryad.280gb5mx0.

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

## Acknowledgements

We thank Dr. Sara Weinstein and all authors of van Schaik et al. 2015 (https://doi.org/10.1371/journal.pone.0130850) for sharing their data on their study taxa through personal communications. This research was supported by the National Science Foundation (IBN-9874523, CNS-025214, and IOB-9874523, DIR) and the Simons Foundation (grant #638529, KJT).

## Author contributions

All authors conceived of the study and developed the study design. KJT and DIR acquired funding for the study. KJT and SBSWH conducted data collection and visualization. KJT wrote the initial draft and all authors reviewed and edited the manuscript.

## Competing interests

The authors declare no competing interests.
