## [Peer Review File · Nature Communications]

New estimates indicate that males are not larger than females in most mammalsReviewers' Comments:

Reviewer #1:

Remarks to the Author:

Dear authors, I love this paper. I think about sex dimorphism a lot and in one of the "untypical" taxa, which you now show to be perfectly typical. Being influenced by the existing literature I had never done the mental switch that you now prompted me to do. The only thing I regret (and I realize that the format you submitted it as does not allow this) is that the paper is too short! I would have loved more predictions and discussion about why males should not always be larger even in species with male competition, more analyses about variation of this within taxa and maybe more about how the results might be different if only those males that get access to females were included, not all adult males. The only thing I am maybe not 100% clear about is the body length vs mass one (and it goes into the direction of the last thing I said). Why would that be sampled differently? Maybe I just did not get that part.

I recommend publication as is.

Reviewer #2:

Remarks to the Author:

I find the manuscript very relevant, neatly done and excellently exposed. I suggest the authors that they review the various articles that I have published on this subject with the hope that they will find in them some argument that contributes to their highly significant discussion on the topic of sexual dimorphism in mammals.

Cassini MH (2017) Role of fecundity selection on the evolution of size sexual dimorphism in mammals. *Animal Behaviour* 128:1-4.

Cassini MH (2020) A mixed model of the evolution of polygyny and sexual size dimorphism in mammals. *Mammal Review* 50: 112-120.

Cassini MH (2020) Sexual size dimorphism and sexual selection in primates *Mammal Review*, 50: 231-239.

Cassini MH (2020) Sexual size dimorphism and sexual selection in Artiodactyls. *Behavioral Ecology*, 31: 792-797.

Cassini, M. H. (2021). Sexual aggression in mammals. *Mammal Review* 51:247-255.

Cassini, M. H. (2022). Evolution of sexual size dimorphism and sexual segregation in artiodactyls: the chicken or the egg?. *Mammalian Biology*, 1-11.

Cassini MH (2023). Measuring sexual selection in mammals. *Hystrix* 33: 123-125.

Marcelo

Reviewer #3:

Remarks to the Author:

This is an interesting and well-written manuscript on the prevalence of dimorphism in mammals. However, its overblown rhetoric is way over the top for a scientific publication. For example, in this publication it is reported (from a sample size of 405 species) that males are larger than females in 44 % of investigated mammal species. For comparison, in Lindenfors et al. 2007 it is reported (from a less carefully chosen sample size of 1370 species) that males are larger than females in 45 % of investigated mammal species. Yet we are supposed to be persuaded that this manuscript overturns common knowledge on mammal dimorphism? And that there is "need to revisit other assumptions in sexual selection research". No. This is just another data-set.

The manuscript deserves to be published, but not in a prestigious general journal but in a more

specialized journal on mammals. There are several to choose from. What is reported is common knowledge among mammalogists, but still interesting. Yes, most mammals are rodents and bats, and the male-biased dimorphism there is not as prevalent (though it is very common in rodents). This has to do with the general small size and airborne life-style of these groups. We know this already. The interest lies more in the updated data-set.

I would advise to tone down the rhetoric several steps. This study does not warrant banging on the loudest drum. This is just bad scientific practice. If there is a political point to be made it should be made in popular journals or in publications specifically on gender studies.

Reviewer #4:

Remarks to the Author:

The primary purpose of your review is to provide feedback on the soundness of the research reported. This will help authors to improve their manuscript and editors to reach a decision. When composing your report, the following questions might assist you in writing a well-justified review, but please feel free to raise any further questions and concerns about the paper.

- What are the noteworthy results?
- Will the work be of significance to the field and related fields? How does it compare to the established literature? If the work is not original, please provide relevant references.
- Does the work support the conclusions and claims, or is additional evidence needed?
- Are there any flaws in the data analysis, interpretation and conclusions? - Do these prohibit publication or require revision?
- Is the methodology sound? Does the work meet the expected standards in your field?
- Is there enough detail provided in the methods for the work to be reproduced?

Dear authors

I have now finished reading your manuscript and am very pleased with the contents. I accepted reviewing this work because I was curious of the findings, and when I first read the title and abstract, prior to accepting, I thought the title was far-fetched and somehow misleading, but after reading the whole rationale given by the authors, I must say I am convinced and changed my mind.

I only have minor comments, some of which may not be implemented, as I am unaware if there are size restrictions for manuscripts. If that is the case, please feel free to use the comments to further explore the data.

First, I am pleased with the review on SSD literature. Authors have done a great job in reviewing that literature, both on the extent and misuses of the actual existence of male biased SSD in mammals, and in retrieving and bringing evidence on the actual problems in defining SSD. More than the raw findings of the extent of male- or female-biased SSD or monomorphism, the discussion of several of the problems (and possible sources for those problems) constitutes, in my view, the main strength of this work.

I was initially worried by the sample size and its representativeness for a group with over 6,000 species, but the logic behind the data inclusion is well explained, and authors do a good job in relating their work with other with larger datasets. With that in mind, they deserve recognition for the rigor applied in data inclusion, even if 5% seems low at first glance.

Authors must also be praised for aiming at deconstruction several biases that have been present, which most likely reflect implicit bias in those who wrote them. I would be interested in, for example, an analysis of gender distribution among authors that support or refuse the idea of a widespread prevalence of male-biased SSD, but that would fall into a whole other scope than this study.

I understand specific discussion on the presence of each type of SSD or its absence in each group may be beyond the goal of this manuscript, but I wonder if it would be at least be possible to comment on the dichotomy (in your data) between the pattern found in placentals and marsupials. While I understand much more variation could be expected in placentals because of the diversity in

morphologies, habits and body sizes, it is striking that all marsupials orders included in the analysis are on the right side of Figure 2 and no marsupial family shows female-biased SSD. Overall, even if these results confirm other findings, its rigorous approach and its discussion are worthy of publication. I am satisfied with the methods and analyses presented .

REVIEWER COMMENTS

Please find our point-by-point responses below in blue font.

Reviewer #1 (Remarks to the Author):

Dear authors, I love this paper. I think about sex dimorphism a lot and in one of the "untypical" taxa, which you now show to be perfectly typical. Being influenced by the existing literature I had never done the mental switch that you now prompted me to do. The only thing I regret (and I realize that the format you submitted it as does not allow this) is that the paper is too short! I would have loved more predictions and discussion about why males should not always be larger even in species with male competition, more analyses about variation of this within taxa and maybe more about how the results might be different if only those males that get access to females were included, not all adult males.

We thank the reviewer for their enthusiasm for the study! We have added a paragraph to the Discussion to flesh out some existing hypotheses on monomorphic and female-biased dimorphic clades, and on alternative reproductive strategies in both males and females – some hypotheses are taxon-specific and others are more generalizable, and we have highlighted areas that could use more theory development.

The only thing I am maybe not 100% clear about is the body length vs mass one (and it goes into the direction of the last thing I said). Why would that be sampled differently? Maybe I just did not get that part.

I recommend publication as is.

Since the body length analysis was a secondary/confirmatory analysis in the paper, we simply gleaned what body length data were available from the sources that we used for body mass, so the body length data are from a more restricted sample of species (N=172 species). This was not enough data to allow us to sample 5% from each order, but we still ran a taxonomically-balanced analysis for body length by sampling 1% of each order (and did so over 1,000 iterations so as to use all of the available data without allowing overrepresented taxa to be unduly weighted in the estimates, just as we did for the body mass data).

Reviewer #2 (Remarks to the Author):

I find the manuscript very relevant, neatly done and excellently exposed. I suggest the authors that they review the various articles that I have published on this subject with the hope that they will find in them some argument that contributes to their highly significant discussion on the topic of sexual dimorphism in mammals.

We thank the reviewer for highlighting this oversight, and we were actually surprised to see that the submitted version did not include a citation from your work as we have been discussing your papers during the writing of this study. We have now included some of these in our manuscript, and are particularly intrigued by your recent analysis finding low variance

in paternity skew among male mammals, which we see as a very significant advancement of sexual selection research.

Cassini MH (2017) Role of fecundity selection on the evolution of size sexual dimorphism in mammals. *Animal Behaviour* 128:1-4.

Cassini MH (2020) A mixed model of the evolution of polygyny and sexual size dimorphism in mammals. *Mammal Review* 50: 112-120.

Cassini MH (2020) Sexual size dimorphism and sexual selection in primates *Mammal Review*, 50: 231-239.

Cassini MH (2020) Sexual size dimorphism and sexual selection in Artiodactyls. *Behavioral Ecology*, 31: 792-797.

Cassini, M. H. (2021). Sexual aggression in mammals. *Mammal Review* 51:247-255.

Cassini, M. H. (2022). Evolution of sexual size dimorphism and sexual segregation in artiodactyls: the chicken or the egg?. *Mammalian Biology*, 1-11.

Cassini MH (2023). Measuring sexual selection in mammals. *Hystrix* 33: 123-125.

Marcelo

Reviewer #3 (Remarks to the Author):

This is an interesting and well-written manuscript on the prevalence of dimorphism in mammals. However, its overblown rhetoric is way over the top for a scientific publication. For example, in this publication it is reported (from a sample size of 405 species) that males are larger than females in 44 % of investigated mammal species. For comparison, in Lindenfors et al. 2007 it is reported (from a less carefully chosen sample size of 1370 species) that males are larger than females in 45 % of investigated mammal species. Yet we are supposed to be persuaded that this manuscript overturns common knowledge on mammal dimorphism? And that there is "need to revisit other assumptions in sexual selection research". No. This is just another data-set.

Our message is that the 'larger males' narrative has remained the prevailing one *despite* counterevidence. We specifically use the Lindenfors study to illustrate this inertia, as well as the need for testing the question using statistical tests rather than only descriptive statistics. Here is a direct quote from their concluding paragraph where they fall back on mean mass ratios to promote the larger males narrative:

'We find that, on average, male mammals are the larger sex (average male/female mass ratio 1.184), with males being at least 10% larger than females in over 45% of species.'

In any case, while the counterevidence so far has been suggestive, *all* evidence so far is based on both crude estimates and taxonomically-biased data. We are the first to show that the larger males narrative is unlikely to be true using statistically-determined dimorphism and a weighting of the estimates by species richness. Further, while we do not wish to diminish the achievements of Ralls (1977) or Lindenfors et al. (2007), we are in fact the first to test this

question and conclude that males are not larger than females in most mammalian species, in part because we were able to do so with statistical rigor. This is overturning common knowledge, as can be seen in the many (including recent) citations that have touted the larger males narrative, and in the surprised reaction from all three of the other reviewers. I have presented our study at several meetings/seminars and at each one the audience, including experts in mammalogy, sexual selection, and animal behavior, have been surprised and have gone on to amend their lectures to undergraduates accordingly. That being said, we do not pretend to have final say over the matter and devote a paragraph in our Discussion to remaining gaps in the data and state of knowledge.

The manuscript deserves to be published, but not in a prestigious general journal but in a more specialized journal on mammals. There are several to choose from. What is reported is common knowledge among mammalogists, but still interesting. Yes, most mammals are rodents and bats, and the male-biased dimorphism there is not as prevalent (though it is very common in rodents). This has to do with the general small size and airborne life-style of these groups. We know this already. The interest lies more in the updated data-set. See above. An additional note on rodents and bats: some of the counterevidence to the larger males narrative so far has come from researchers focusing on these clades (e.g., Lu et al. 2014), but we are still the first to test the question statistically across mammals, weighting each clade by their species richness and giving these speciose clades of small mammals the emphasis they deserve. In other words, it has been known that female-biased size dimorphism is common in bats and that there is a lot of monomorphism in rodents (Ralls 1977 points this out), but how that affects the 'norm' in mammals more generally has not been tested other than with the crude and taxonomically-biased analyses we mention above, which have promoted the 'larger males' narrative.

I would advise to tone down the rhetoric several steps. This study does not warrant banging on the loudest drum. This is just bad scientific practice. If there is a political point to be made it should be made in popular journals or in publications specifically on gender studies. We have gone through the manuscript to make the language more toned down without diluting the message. However, we believe it is critical to recognize the biases that may underlie our science (and the male focus in sexual selection research is well-documented, see our citations), as this is important for responsible scientific practice and for advancing our field.

Reviewer #4 (Remarks to the Author):

Dear authors

I have now finished reading your manuscript and am very pleased with the contents. I accepted reviewing this work because I was curious of the findings, and when I first read the title and abstract, prior to accepting, I thought the title was far-fetched and somehow misleading, but after reading the whole rationale given by the authors, I must say I am convinced and changed my mind.

I only have minor comments, some of which may not be implemented, as I am unaware if there are size restrictions for manuscripts. If that is the case, please feel free to use the comments to further explore the data.

First, I am pleased with the review on SSD literature. Authors have done a great job in reviewing that literature, both on the extent and misuses of the actual existence of male biased SSD in mammals, and in retrieving and bringing evidence on the actual problems in defining SSD. More than the raw findings of the extent of male- or female-biased SSD or monomorphism, the discussion of several of the problems (and possible sources for those problems) constitutes, in my view, the main strength of this work.

I was initially worried by the sample size and its representativeness for a group with over 6,000 species, but the logic behind the data inclusion is well explained, and authors do a good job in relating their work with other with larger datasets. With that in mind, they deserve recognition for the rigor applied in data inclusion, even if 5% seems low at first glance.

Authors must also be praised for aiming at deconstruction several biases that have been present, which most likely reflect implicit bias in those who wrote them. I would be interested in, for example, an analysis of gender distribution among authors that support or refuse the idea of a widespread prevalence of male-biased SSD, but that would fall into a whole other scope than this study.

I understand specific discussion on the presence of each type of SSD or its absence in each group may be beyond the goal of this manuscript, but I wonder if it would be at least possible to comment on the dichotomy (in your data) between the pattern found in placentals and marsupials. While I understand much more variation could be expected in placentals because of the diversity in morphologies, habits and body sizes, it is striking that all marsupials orders included in the analysis are on the right side of Figure 2 and no marsupial family shows female-biased SSD.

Overall, even if these results confirm other findings, its rigorous approach and its discussion are worthy of publication. I am satisfied with the methods and analyses presented.

Thank you very much for this encouraging feedback! Please see our responses below, which also address points you raise above.

Points raised directly in-text in the attachment:

-Title: Even though the estimates are new, authors could consider using “confirm”, “reinforce” or some other wording indicating that the results are not NOVEL even if the analysis or approach is.

We have given this some thought. The difficulty is that while the results (specifically the rate of male-biased dimorphism) has been suggested as a possibility (Ralls1977) and has been implicit in the results of one other study before (Lindenfors et al. 2007), the interpretation/conclusion that most mammals do not have larger males *is* novel. We feel that our emphasis on a rigorous estimate of rates and on challenging the 'larger males' narrative does in fact provide novel results and interpretation and we would like to keep the title as is, expressing that these are new, updated estimates and that they suggest most mammals do not have larger males.

-Line 31: I do not have access to several of these sources, but those I could read sometimes do state that “males are typically larger than females”, but others, such as the review by Abouheif and Fairbairn state mostly that WHEN there is SSD in mammals, it is male biased.

We are not so sure. See the quote in context:

Sexual differences in size and morphology are widespread in the animal kingdom. In most species of animals, females attain larger body sizes than do males (e.g., most spiders, insects, fish, amphibians, reptiles), whereas in most birds and mammals, males are the larger sex (Darwin 1874; Selander 1972; Ghiselin 1974; Ralls 1977; Alexander et al. 1979; Greenwood and Wheeler 1983; Arak 1988; Lewin 1988; Shine 1988; Hedrick and Temeles 1989).

However, if you are taking the introductory sentence to imply that only size-dimorphic species are treated in the following sentence, others may as well, and we have removed this citation and any others that used a similar preceding sentence to make sure we are giving equal benefit of the doubt to all sources. We found a couple of other sources promoting the ‘larger males’ narrative, and so the list of clear-cut statements supporting the narrative is now: Trivers 1972, Greenwood and Wheeler 1985, Dinerstein 2003, Lindenfors et al. 2007, Cassini 2017, and Mori et al. 2017. We are happy to provide any sources to which you do not have access if you wish.

-Line 125: I understand orders are listed alphabetically but this masks in part the difference between marsupials and placentals. I suggest grouping the former and the latter and then list in alphabetical order.

This is a great idea. We went ahead and reordered the whole y axis according to relatedness and attached a phylogeny on the left side so that readers can see how the various higher-order clades compare on the dimorphism spectrum, including placentals vs. marsupials.

-Figure 2: These arrows could be placed below the graph rather than inside as they overlap with the Scandentia data.

Good idea. Done.

-Line 190: While I agree that body mass has some problems because of condition, it is also possible that condition may be the character that is being dimorphic. I think body mass may have more biological meaning (or, at least, interpretable meaning) than length, especially across a group so diverse in morphology and body types as mammals. Mass may reflect strength and advantage in those species with male-biased SSD, may represent the investment or ability of females to be better prepared for enduring pregnancy and maternal care in those species with female-biased SSD, or be interpreted as several sorts of trade-offs in monomorphic species. Obviously, if these data could be compared to mating system, then we could have a better picture. Unfortunately, I am almost certain that for most of these smaller and monomorphic taxa, mating system may be simply unknown...

Yes, all in all we agree that body mass is a good measure for the focus of this paper, as it is more amenable to broad comparisons across taxa than body length and may give an individual a physical advantage in multiple ways. We do note that for many small mammals,

and for bats in particular, experts strongly prefer to use body length because of the rapid fluctuations in mass individuals can experience even within a day. However, for our study it was logical to use mass as it is the most widely available measure of body size across the literature, is measured in a more standardized way than body length, and has perhaps more generalizable interpretability ('biological meaning') than body length across taxa. We agree with the reviewer that mating system would be a fascinating factor to explore in relation to these data, and are considering it for future studies.

-Lines 286-288: I would be curious to know if the results hold if you use these 609 even with inflation of CI. Could this be done and added?

The results including all 609 species push the rates of monomorphism up to above 50%, but this is not surprising given that smaller sample sizes increase uncertainty and the probability of CI overlap between females and males and therefore the rate of monomorphism (see Fig. S1). Given our emphasis on quality data and rigorous analysis as one of the main contributions of this paper, we prefer not to add analyses including lower-quality data as they do not add any clarity to the story.

-Line 326: I know this is not cited in the text, but you leave the reader curious here. If not listing the reference, then provide the DOI here?

Yes, good idea!

Reviewers' Comments:

Reviewer #1:

Remarks to the Author:

I love this paper. I think about sex dimorphism a lot and in one of the "untypical" taxa, which you now show to be perfectly typical. Being influenced by the existing literature I had never done the mental switch that you now prompted me to do. The only thing I regret (and I realize that the format you submitted it as does not allow this) is that the paper is too short! Thank you for incorporating this more in the discussion. Looking forward to seeing it out.

Dina Dechmann

Reviewer #4:

Remarks to the Author:

I have read your responses to my comments, and I am pleased to inform that I feel they address all my concerns.

Congratulations on a very interesting analysis.